# Augmenting Zero-Shot Dense Retrievers with Plug-in Mixture-of-Memories

**Suyu Ge**[1*], **Chenyan Xiong**[2*], **Corby Rosset**[3], **Arnold Overwijk**[3]
**Jiawei Han**[1], **Paul Bennett**[4*]

[1] University of Illinois Urbana-Champaign   [2] Carnegie Mellon University
[3] Microsoft Research   [4] Spotify

{suyuge2,hanj}@illinois.edu
cx@cs.cmu.edu
{corbyrosset,arnold.overwijk}@microsoft.com
pbennett@spotify.com

## Abstract

In this paper we improve the zero-shot generalization ability of language models via Mixture-Of-Memory Augmentation (MoMA), a mechanism that retrieves augmentation documents from multiple information corpora ("external memories"), with the option to "plug in" unseen memory at inference time. We develop a joint learning mechanism that trains the augmentation component with latent labels derived from the end retrieval task, paired with hard negatives from the memory mixture. We instantiate the model in a zero-shot dense retrieval setting by augmenting strong T5-based retrievers with MoMA. With only T5-base, our model obtains strong zero-shot retrieval accuracy on the eighteen tasks included in the standard BEIR benchmark, outperforming some systems with larger model sizes. As a plug-in-play model, our model can efficiently generalize to any unseen corpus, meanwhile achieving comparable or even better performance than methods relying on target-specific pretraining. Our analysis further illustrates the necessity of augmenting with mixture-of-memory for robust generalization, the benefits of augmentation learning, and how MoMA utilizes the plug-in memory at inference time without changing its parameters. Our code can be found at https://github.com/gesy17/MoMA.

## 1 Introduction

Scaling up language models—with more parameters and pretraining data—improves model generalization ability on downstream applications (Raffel et al., 2019; Brown et al., 2020; Smith et al., 2022), but with diminishing return: *linear* improvements on downstream metrics often require *exponentially* more parameters and computing cost (Kaplan et al., 2020; Hoffmann et al., 2022). Hence, scaling pretrained language models in this way is economically unsustainable (Strubell et al., 2020; Bender et al., 2021; Zhang et al., 2022).

Retrieval augmented language models provide a promising alternative. They allow language models to efficiently access vast resources from an external corpus (Guu et al., 2020; Borgeaud et al., 2022) that serves as a kind of "memory" they can refer to when making predictions, alleviating the need to memorize as much information in their own network parameters (Roberts et al., 2020). This open-book approach helps language models to better generalize on token prediction tasks and machine translation (Khandelwal et al., 2019; Borgeaud et al., 2022), and tasks which already involve a first-stage retrieval component, e.g., OpenQA (Borgeaud et al., 2022; Izacard et al., 2022). Existing retrieval augmentation methods usually stick to **one** single retrieval corpus throughout training and inference so that the retrieval component can be indirectly guided by the supervision from end tasks.

In this paper we improve the zero-shot generalization ability of language models using "mixture-of-memory" (MoMA), a new retrieval augmentation mechanism. Instead of a single corpus, MoMA retrieves documents from a "mixture" of multiple external corpora and enjoys the merits of a larger and more comprehensive source of knowledge. This mechanism also allows removing and/or "plugging-in" new corpora during inference time, when more information from the target task is revealed, or as an additional way for users to control the model. Specifically, we apply MoMA on the zero-shot dense retrieval task, which is the foundation of many important real-world applications (Thakur et al., 2021a; Kim, 2022) and also the retrieval component of recent retrieval augmented language models (Guu et al., 2020; Izacard et al., 2022). However, it is not trivial to guide a retrieval model to leverage multiple corpora. We need to jointly train the augmentation component and dense retriever using supervised relevance signals and self-mined hard negatives.

---

*Work partly done while at Microsoft.

We instantiate MoMA with a T5-base model (Ni et al., 2022) and apply it to the dense retrieval task (Karpukhin et al., 2020). Our end task retriever uses a set of augmenting documents from the mixture-of-memories to enhance its representation of the query with important context; the retriever then uses the enhanced query representation to retrieve a final candidate set. At inference time, we plug in the target task's corpus to the memory mixture to introduce in-domain context information, without updating any parameter.

As a plug-in-play method, MoMA provides an flexible but powerful solution to zero-shot dense retrieval: Unlike recent state-of-the-art methods (Yu et al., 2022; Neelakantan et al., 2022), it does not require pretraining on target corpus or large-scale web corpus, enabling it to generalize to arbitrary unseen corpus without additional effort. It can also be used as an efficient alternative for recent large language model (LLM) based generative retrieval models (Gao et al., 2022). Given the target query, MoMA only involves the T5-base model for query encoding, which is significantly cheaper than querying an LLM to generate pseudo answers and re-encoding it.

We experimented on eighteen zero-shot dense retrieval tasks included in BEIR (Thakur et al., 2021a), the standard ZeroDR benchmark. The results demonstrate the improved zero-shot ability of MoMA. MoMA achieves comparable or even stronger performance to recent state-of-the-art dense retrieval systems with larger model scales and heavier computation costs. Our further analysis reveals that large and diverse corpora in the memory leads to the best performance; while only using a single corpus during training does not improve performance on unseen target tasks. The learning of augmentation component is also important for MoMA to utilize the diverse information from the mixture. Our analysis and case studies illustrate how MoMA leverages the plug-in memory at testing time to enrich its query representations.

## 2 Related Work

### 2.1 Retrieval Augmentation

Recent research has explored the retrieval-augmented language model, which aims to construct an external memory for the language model (Khandelwal et al., 2019; Zhong et al., 2022; Guu et al., 2020; Borgeaud et al., 2022; Petroni et al., 2020). It retrieves related documents or to-

kens from an in-domain corpus as additional input to enhance the semantic representation. Despite their effectiveness, learning to retrieve useful documents to augment the language model is a challenging task, since human annotations on the usefulness of augmentation documents are costly and seldom available. The most straightforward way is to use representations from raw pretrained language models, i.e., as unsupervised dense retrieval (Guu et al., 2020; Borgeaud et al., 2022). Adapting existing dense retrieval models is another common choice (Izacard and Grave, 2020b; Lewis et al., 2020; Yu et al., 2021). A more plausible solution is to jointly learn the augmentation components end-to-end using supervision from the final task, for example, treating the augmentation as latent variables and applying EM (Zhao et al., 2021), or distilling the augmentation component from feedback of the final model (Izacard and Grave, 2020a). In a parallel work, Izacard et al. (2022) found the most effective one is attention distillation method (ADist), which trains the augmentation component using soft labels derived from the end model's attention on augmentation documents.

The motivation for query augmentation coincides with the query expansion methods in the traditional IR community, whereby a query is augmented by new but similar features (Carpineto and Romano, 2012). As feature selection usually requires additional semantic analysis, the efficiency and usability of traditional query expansion methods remain limited when faced with a new domain. To overcome this, recent work relies on dense retrieval results to expand the query (Yu et al., 2021). The retrieved relevant documents serve as pseudo relevance feedback signals for the model, which are concatenated with the original query as the augmented model input. Our work augments queries with feedback from multiple corpora and learns to select important augmentation documents automatically.

### 2.2 Zero-shot Dense Retrieval

Dense retrieval models trained on a resource rich source tasks, e.g., web search, usually do not perform as well when zero-shot transferred to other domains (Thakur et al., 2021b). Xin et al. (2021) analyzed the challenge of shifting between training and testing domains, and leveraged domain-invariant learning to mitigate the gap. Another common approach is to first generate domain-specific pseudo

labels for each task, and then use them to train dense retriever (Thakur et al., 2021b; Wang et al., 2022). Additionally, continuous pretraining the language model also improves its generalization ability in ZeroDR (Izacard et al., 2021). Following works (Izacard et al., 2021; Yu et al., 2022) further contrastively pretrained the retriever on target or external corpus with a sentence matching loss. One significant drawback of them is requiring the target or external corpus as part of the training corpus, which prohibits the plug-in-play feature when exposed to new data. Besides, stacking all target datasets for model pretraining also increases computation costs to a notable degree. On BEIR benchmarks which contain 18 target tasks, it enlarges the training corpus to 7 times larger.

Other methods seek better generalization ability in ZeroDR from various resources, for example, combining with sparse retrieval to introduce exact match signals (Formal et al., 2021) or using multiple vectors per documents for term-level matching (Khattab and Zaharia, 2020a). More recent work simply changes backbone models to larger language models, such as T5-XXL or cpt-text-XL (Ni et al., 2021; Neelakantan et al., 2022). Some rely on stronger instruction-guided generative language models (Gao et al., 2022), which match documents with model-generated query answers. Overall, methods relying on large language models will incur heavier costs on memory consumption and computation, and calling generative model API may also cause latency issues. Instead of chasing stronger backbone models, our goal in this paper is to provide an efficient plug-in-play alternative for them.

## 3 Method

In this section we first describe our Mixture-of-Memory Augmentation. Then we discuss how it is jointly learned with the end system and enables plug-in memory at inference time.

### 3.1 Mixture-of-Memory Augmentation

Before going to the details of MoMA, we first recap some preliminaries in ZeroDR.

**Preliminaries.** The dense retrieval (DR) task aims to find relevant documents $d$ from a corpus $C$ for the given query $q$ by representing them in a shared embedding space. Specifically, the retrieval score in DR is often calculated as:

$$f(q, d) = \boldsymbol{q} \cdot \boldsymbol{d}; \boldsymbol{q} = g(q); \boldsymbol{d} = g(d). \quad (1)$$

It uses dot product as the scoring function to match the embeddings $\boldsymbol{q}$ and $\boldsymbol{d}$, which is known to support efficient approximate nearest neighbor search (ANN) (Johnson et al., 2019). A pretrained language model is often the encoder of choice $g()$. We use the ST5-EncDec variant of Sentence-T5 (Ni et al., 2022):

$$g(x) = \text{Dec}(\text{Enc}(x)), \quad (2)$$

which feeds in the text sequence (prepended by a special [CLS] tokens) to the encoder of T5, $\text{Enc}()$, and uses the output representation of the [CLS] token from the decoder, $\text{Dec}()$, as the text representation. This naturally leverages the attention from decoder to encoder at all Transformer layers (Raffel et al., 2019), as a fine-grained information gathering mechanism.

The *training* of dense retrieval systems often applies standard ranking loss and pairs the relevant documents $d^+ \in D^+$ for each query q with hard negatives $d^- \in D^-$:

$$\mathcal{L} = \sum_q \sum_{d^+ \in D^+} \sum_{d^- \in D^-} l(f(q, d^+), f(q, d^-));$$
$$D^- \sim \text{ANN}_{f(q, \circ)}^C \setminus D^+. \quad (3)$$

Eqn. 3 uses ANCE hard negatives, which are the top-retrieved documents from $C$ using the retriever itself (Xiong et al., 2020). The loss function $l()$ can be any standard ranking loss such as cross entropy. A ZeroDR model is trained on $q^s$ and documents $d^s \in C^s$ from a *source task*, often web search, and tested on *target* tasks $q^t$ and $C^t$; supervision signals are only present from the source.

**Mixture-of-Memory.** The key idea of (document-based) retrieval augmented language models is to enrich the representation $g(q)$ with additional contextual input for the model, i.e., augmentation documents $d^a$ retrieved from an external memory $\mathcal{M}$. Instead of using a single document corpus, MoMA uses multiple corpora to provide richer and more diverse external resources for augmentation. For example, $\mathcal{M}$ can be composed by the source corpus $C^s$, a general encyclopedia, a domain specific knowledge graph, etc. Then we can retrieve the augmentation documents $D^a$ :

$$D^a = \text{ANN}_{f^a(x, \circ)}^{\mathcal{M}}; \mathcal{M} = \{C_1, ..., C_M\}. \quad (4)$$

This augmentation component uses another dense retriever $f^a()$, which also adopts the Sentence-T5 architecture. Note that instead of retrieving $D^a$

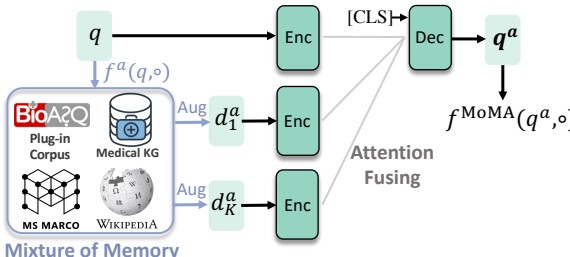

Figure 1: Illustraion of the Mixture-of-Memory Augmentation.

separately from $M$ different ANN memory sources and merging results, Eqn. 4 combines them into one ANN index. This requires the augmentation component $f^a()$ to be flexible enough to handle various corpora in the mixture.

Using the encoder-decoder architecture for $g()$ in Eqn. 2 enables a simple extension to incorporate the augmentation documents using the fusion-in-decoder (FiD) mechanism (Izacard and Grave, 2020b):

$$g^{\text{MoMA}}(q) = \text{Dec}(\text{Enc}(q), \text{Enc}(d_1^a), ..., \text{Enc}(d_K^a));$$
$$D^a = \{d_1^a, ..., d_K^a\}. \qquad (5)$$

It feeds in the $K$ augmentation documents separately to the T5 encoder of $g()$. Then it fuses the encoded documents together with $\text{Enc}(q)$ using one decoder that attends to all encoded vectors, as illustrated in Figure 1.

The FiD approach in Eqn 5 is a nice balance of efficiency and capacity when modeling multiple text sequences (Izacard and Grave, 2020b). It is more efficient than concatenating all text pieces together, while also remaining expressive enough to model the nuances from many sequences. (Izacard and Grave, 2020a; Izacard et al., 2022).

When instantiating MoMA in the dense retrieval setting, we focus on augmenting the query representation $\boldsymbol{q}$, as queries are often short, ambiguous, and benefit more from additional contextual information (Lavrenko and Croft, 2017; Yu et al., 2021). This leads to the following definition of MoMA:

$$f^{\text{MoMA}}(q, d) = \boldsymbol{q}^a \cdot \boldsymbol{d};$$
$$\boldsymbol{q}^a = g^{\text{MoMA}}(q), \boldsymbol{d} = g(d), \qquad (6)$$

using the construction of $g^{\text{MoMA}}()$ in Eqn. 5 upon the augmentation documents defined in Eqn. 4.

## 3.2 Joint Learning in MoMA and Inference with Plug In Memory

MoMA has two sets of parameters to learn, in the main model $f^{\text{MoMA}}()$ and the augmentation component $f^a()$. Both have their own independent parameters. The two components are bridged by the augmentation documents, which are retrieved by $f^a()$ from $\mathcal{M}$ and used by $f^{\text{MoMA}}()$ to produce query representation $\boldsymbol{q}^a$.

**Main Model Learning.** Given the relevance labels from the source task and an augmentation model, training $f^{\text{MoMA}}()$ is straightforward. We can use the standard dense retrieval training to fine-tune the enriched query encoder $g^{\text{MoMA}}()$ and the document encoder $g()$:

$$\mathcal{L}^{\text{MoMA}} = \sum_{q^s, d^+, d^-} l(f^{\text{MoMA}}(q^s, d^+), f^{\text{MoMA}}(q^s, d^-));$$
$$d^+ \in D^{s+}, d^- \in D^{s-} \qquad (7)$$
$$D^{s-} \sim \text{ANN}_{f^{\text{MoMA}}(q^s, \circ)}^{C^s} \setminus D^{s+}. \qquad (8)$$

The training signals come from the source task, including $q^s$, its relevant documents $D^{s+}$, and ANCE hard negatives $D^{s-}$ retrieved from the source corpus $C^s$.

**Augmentation Learning.** Training $f^a()$ is challenging as it is hard to label whether an augmentation document is useful. Propagating gradients from the final loss to $f^a()$ is also prohibitive as the retrieval operation in Eqn. 4 is discrete. Fortunately, recent research found the attention scores from the FiD decoder to each encoded inputs (Eqn. 5) are good approximations to the usefulness of augmentation documents (Izacard and Grave, 2020a):

$$\text{FidAtt}(d_i^a) = \sum_{\text{pos}} \sum_{\text{head}} \text{Att}_{\text{Dec} \to \text{Enc}}(g^{\text{MoMA}}(d_i^a)). \qquad (9)$$

It sums the attentions from $g^{\text{MoMA}}()$'s special token at the decoder's [CLS] position over all layers, input positions, and attention heads. Ideally, higher FidAtt() is assigned to $d_i^a$ that provides useful contextual information.

Previously, FidAtt scores are often used as soft labels for the augmentation model (Izacard and Grave, 2020a; Izacard et al., 2022). Doing so with memory mixtures is risky as it is too sparse and overfits memory resource that appears earlier in the training, which are the only ones available for the decoder to attend on. To improve the learning robustness, we introduce ANCE-style hard negative mining to train the augmentation component as well.

First, we formulate the positive set of augmenta-

tion documents as:

$$D^{a+} = D^{s+} \cup \text{Top-N}_{\text{FidAtt}(d_i^a), D^a}. \qquad (10)$$

which combines relevant documents $D^{s+}$ and the augmenting ones that received N-highest attention scores from $g^{\text{MoMA}}()$. Then we pair them with hard negatives to formulate the training of $f^a()$ as:

$$\mathcal{L}^a = \sum_{q^s} \sum_{d^+ \in D^{a+}} \sum_{d^- \in D^{a-}} l(f^a(q^s, d^+), f^a(q^s, d^-));$$
$$\qquad (11)$$

$$D^{a-} \sim \text{ANN}_{f^a(q^s, \circ)}^{\mathcal{M}} \setminus D^{a+}. \qquad (12)$$

Notice the negatives for $f^a()$ have comprehensive coverage from multiple corpora.

**Iterative Training.** The learning of $f^{\text{MoMA}}()$ and $f^a()$ is an iterative process that fits naturally into the training procedure of dense retrieval training with hard negatives. We follow the standard iterations in ANCE and construct the $t$-th training episode of MoMA:

1. Construct hard negatives $D^{s-}$ via Eqn. 8 using weights $f_{t-1}^{\text{MoMA}}()$ from the last episode;

2. Retrieve augmentation $D^a$ via Eqn. 4 using weights $f_{t-1}^a()$ from the last episode;

3. Train $f_t^{\text{MoMA}}()$ as Eqn. 7;

4. Formulate new positive augmentation documents $D^{a+}$, using updated attention scores from $f_t^{\text{MoMA}}()$, and mine negative augmentation documents $D^{a-}$ using $f_{t-1}^a()$;

5. Train $f_t^a()$ following Eqn. 11.

Both $f_0^{\text{MoMA}}()$ and $f_0^a()$ can be initialized with a BM25 warmed-up T5 retriever. Steps 1 and 3 above are inherited from standard dense retrieval training. The rest are introduced by MoMA. The additional computation in the training side mainly resides updating the index for the memory mixture, a standard cost in retrieval-augmented language models (Guu et al., 2020; Izacard et al., 2022).

**Zero-Shot Retrieval with Plug in Memories.** To perform zero-shot retrieval on unseen tasks, MoMA first retrieves augmented documents using $f^a()$ from $\mathcal{M}$ for the target query $q^t$, and retrieves target documents $d^t \in C^t$ with the augmented model $f^{\text{MoMA}}()$ without changing any model parameters. MoMA allows $f^a()$ to attend over the target corpus as well if it is plugged in: $\mathcal{M} =$ $\mathcal{M} \cup C^t \setminus C^s$, which conveys in-domain information. The augmenting corpus can also be engineered by users manually to inject their preference or domain knowledge, e.g., as "memory engineering". In this work we focus on swapping out the source corpus for the target corpus; we leave other explorations for future work.

## 4 Experimental Methodologies

**Datasets.** We choose the MS MARCO passage dataset (Bajaj et al., 2016) as the source domain dataset, whereas the target domains are from the 18 datasets in BEIR (Thakur et al., 2021b) benchmark, which include including biomedical, scientific and financial texts. More details can be found in Appendix A.3. The evaluation metric NDCG@10 is the same with BEIR benchmark, which measures Normalized Discounted Cumulative Gain (Wang et al., 2013) of top 10 prediction. The higher NDCG@10 value indicates better performance.

**Augmenting Corpora.** During training, the mixture-of-memory is composed of source training corpus (MARCO), Wikipedia and a medical knowledge graph. We use the Wikipedia chunk prepossessed by (Karpukhin et al., 2020) without further processing[1]. The medical knowledge graph is extracted from the Medical Subject Headings (MeSH)[2], an open-source database for indexing and cataloging of biomedical and health-related information. Since it is hierarchical in structure, we linearize it by concatenating spans with text information. During testing, we directly replace MARCO with the corresponding document sets from BEIR. Each task from BEIR is augmented independently. More dataset and preprocessing details can be found in Appendix A.3.

**Baselines and Model Choices.** We compare our MoMA with standard sparse and dense retrieval models on BEIR. We also compare MoMA with advanced approaches that are specifically designed for zero-shot generalization. They involve techniques that are not directly comparable with this paper, including pretraining on extra data, in-domain continuous pretraining, and generating target pairs using another pretrained generative model. Besides, some baselines use larger scale language model as their backbone. We list the details of baselines in Appendix A.4.

As a plug-in-and-play method, MoMA can be

---
[1] https://huggingface.co/datasets/wiki_dpr
[2] https://www.ncbi.nlm.nih.gov/mesh/

Table 1: NDCG@10 on the BEIR benchmark. We also include an averaged score on datasets used by Contriever for a fair comparison. The best result each task is marked bold. An * denotes unfair comparison, as NQ is used in training for GTR. †: GenQ generated pseudo labels to train an independent model for each task. ‡: Larger models

| | BM25 | DPR | ANCE | T5-ANCE | coCondenser | GenQ† | ColBERT | Contriever | GTR$_{base}$* | GTR$_{large}$*‡ | MoMA (T5-ANCE) | MoMA (coCondenser) |
|---|---|---|---|---|---|---|---|---|---|---|---|---|
| **Parameters#** | — | 110M | 110M | 110M*2 | 110M | 66M*18 | 110M | 110M | 110M | 335M | 110M*2 | 110M*2 |
| TREC-COVID | 0.656 | 0.575 | 0.654 | 0.653 | 0.715 | 0.619 | 0.677 | 0.596 | 0.539 | 0.557 | **0.762** | 0.761 |
| BioASQ | 0.465 | 0.232 | 0.306 | 0.322 | 0.318 | 0.398 | **0.474** | — | 0.271 | 0.320 | 0.372 | 0.371 |
| NFCorpus | 0.325 | 0.210 | 0.237 | 0.275 | 0.307 | 0.319 | 0.305 | 0.328 | 0.308 | 0.329 | 0.307 | **0.333** |
| NQ | 0.329 | 0.398 | 0.446 | 0.452 | 0.494 | 0.358 | 0.524 | 0.498 | 0.495 | **0.547** | 0.490 | 0.544 |
| HotpotQA | 0.603 | 0.371 | 0.456 | 0.487 | 0.566 | 0.534 | 0.593 | **0.638** | 0.535 | 0.579 | 0.539 | 0.589 |
| FiQA-2018 | 0.236 | 0.274 | 0.295 | 0.294 | 0.285 | 0.308 | 0.317 | 0.329 | 0.349 | **0.424** | 0.320 | 0.329 |
| Signal-1M | **0.330** | 0.238 | 0.249 | 0.246 | 0.274 | 0.281 | 0.274 | — | 0.261 | 0.265 | 0.258 | 0.264 |
| TREC-NEWS | 0.398 | 0.366 | 0.382 | 0.379 | 0.389 | 0.396 | 0.393 | — | 0.337 | 0.343 | 0.413 | **0.453** |
| Robust04 | 0.408 | 0.344 | 0.392 | 0.412 | 0.399 | 0.362 | 0.391 | — | 0.437 | 0.470 | 0.469 | **0.475** |
| ArguAna | 0.414 | 0.414 | 0.415 | 0.415 | 0.411 | 0.493 | 0.233 | 0.446 | 0.511 | **0.525** | 0.438 | 0.463 |
| Touché-2020 | **0.367** | 0.208 | 0.240 | 0.312 | 0.190 | 0.182 | 0.202 | 0.230 | 0.205 | 0.219 | 0.271 | 0.299 |
| Quora | 0.789 | 0.842 | 0.852 | 0.836 | 0.863 | 0.830 | 0.854 | 0.865 | 0.881 | **0.890** | 0.847 | 0.843 |
| DBPedia-entity | 0.313 | 0.236 | 0.281 | 0.290 | 0.356 | 0.328 | 0.392 | **0.413** | 0.347 | 0.391 | 0.347 | 0.383 |
| SCIDOCS | 0.158 | 0.107 | 0.122 | 0.115 | 0.140 | 0.143 | 0.145 | **0.165** | 0.149 | 0.158 | 0.143 | 0.145 |
| Fever | 0.753 | 0.589 | 0.669 | 0.655 | 0.678 | 0.669 | **0.771** | 0.758 | 0.660 | 0.712 | 0.723 | 0.745 |
| Climate-Fever | 0.213 | 0.176 | 0.198 | 0.194 | 0.184 | 0.175 | 0.184 | 0.237 | 0.241 | **0.262** | 0.235 | 0.233 |
| SciFact | 0.665 | 0.475 | 0.507 | 0.566 | 0.600 | 0.644 | 0.671 | **0.677** | 0.600 | 0.639 | 0.632 | 0.630 |
| CQADupStack | 0.299 | 0.281 | 0.296 | 0.283 | 0.330 | 0.347 | 0.350 | 0.345 | 0.357 | **0.384** | 0.283 | 0.294 |
| Contriever Sub Avg | 0.437 | 0.368 | 0.408 | 0.416 | 0.438 | 0.425 | 0.445 | 0.466 | 0.442 | 0.471 | 0.453 | **0.471** |
| Avg | 0.428 | 0.352 | 0.391 | 0.399 | 0.417 | 0.410 | 0.431 | — | 0.416 | 0.444 | 0.436 | **0.453** |

combined with other techniques. We initiate MoMA on two dense retrieval models. The primitive **MoMA (T5-ANCE)** is built on the original T5 model checkpoint and optimized iteratively with ANCE-style (Xiong et al., 2020) hard negatives. By comparing it with T5-ANCE, we can clearly observe the performance gain brought by MoMA. To demonstrate it can integrate techniques from other models to achieve higher performances, we initiate MoMA on a better pretrained model. Following coCondenser (Gao and Callan, 2022), we continuously trained the original T5 model on the MARCO document corpus using a sentence-level contrastive loss, combined with the original masked language modeling loss. We then performed the same MoMA training on top of the continuously pretrained T5 checkpoint and denoted it as **MoMA (coCondenser)**. The only difference between MoMA (T5-ANCE) and MoMA (coCondenser) is the initialized model start point. We compare their pretraining details with other models in Table 2. Unlike other work (Yu et al., 2022), as a plug-in-play design, we did not include target datasets and augmenting corpora in the contrastive pretraining stage. Since MARCO contains only 0.5M documents, it adds fewer computational overhead compared to other methods listed in the table, e.g., Contriever.

**Implementation Details.** For MoMA, we use the T5-base (Raffel et al., 2019) architecture (12-layer Transformer, 768 hidden size) by directly loading the checkpoint from HuggingFace[3]. To warm up the language model for dense retrieval, we

[3]https://huggingface.co/t5-base

followed (Xiong et al., 2020) to further train it using BM25 negatives for 10 epochs. After warming up, we jointly trained the two components for three episodes, each episode including three training epochs. After three joint episodes, the end retriever reaches the best performance on MSMARCO, so we select this checkpoint for evaluation. The ratio between positive and hard negative pairs is 1:7 for both models. The main hyperparameters in MoMA include the total number of grounding documents $K$ and the attention threshold number N in Equation 10. We directly set $K$=10 and N=5 without any parameter tuning. More details on hyperparameters and experimental settings can be found in Appendix A.5.

## 5 Evaluation Results

### 5.1 Zero-Shot Retrieval Accuracy and Efficiency

The retrieval accuracy of MoMA and baselines are listed in Table 1. Besides baselines of similar parameter count, we also include larger models (GTR$_{large}$) or those using multiple vectors per document (ColBERT). MoMA (coCondenser) shows the strongest zero-shot accuracy against previous state-of-the-art methods that do continuous contrastive pretraining (coCondenser), generate pseudo labels (GenQ), or consume additional training signals in both continuous pretraining and finetuning phrases (GTR$_{base}$). MoMA (T5-ANCE) also achieved nearly comparable zero-shot accuracy against larger models like GTR$_{large}$, and ColBERT, which scales up the number of vectors per documents (one per token). This confirms that retrieval-

Table 2: Computational analysis in the pretraining stage of different models.

| Model | Pretraining Corpus | Batch Size | Training Steps |
|---|---|---|---|
| MoMA (T5-ANCE) | 0 | 0 | 0 |
| MoMA (coCondenser) | MARCO | 128 | 50k |
| GTR$_{base}$ | NQ, CQA | 2048 | 800k |
| Contriever | CCNet | 2048 | 500k |
| | Wikipedia | 2048 | 200k |

Table 3: Efficiency of MoMA search and training.

| Operation | Offline | Online |
|---|---|---|
| BM25 Index Build | 1.8h | — |
| BM25 Retrieval Per Query | — | 43ms |
| **MoMA Inference** | | |
| Encoding of Corpus/Per Doc | 1.5h/4.5ms | — |
| Query Encoding | — | 55ms |
| ANN Retrieval (batched q) | — | 9ms |
| Dense Retrieval Total | — | 64ms |
| **MoMA Training** | | |
| Encoding of Corpus/Per Doc | 1.5h/4.5ms | — |
| ANN Index Build | 10s | — |
| Neg Construction Per Batch (32 queries) | 45ms | — |
| Back Propagation Per Batch (32 queries) | 330ms | — |

augmentation provides another path to improve language models' generalization ability besides scaling up. MoMA (T5-ANCE) also outperforms T5-ANCE, which MoMA (T5-ANCE) uses as a subroutine for retrieval augmentation, on all but one retrieval task, showing the improved generalization ability from plug-in mixture of memory.

We evaluate the efficiency of MoMA in two stages: offline model training and online inference. In offline training from Table 2, MoMA (T5-ANCE) is **significantly cheaper** than other methods as we do not require pretraining on large external corpora, which saves hundreds of hours training time. MoMA (condenser) additionally pretrain on MARCO for 50k steps, which is far fewer than the other compared methods. In online inference, similar with other retrieval enhanced language models, MoMA imposes a necessary cost of retrieval augmented model upon the baseline T5-ANCE. We further provide detailed efficiency analysis on MoMA in Table 3. The online latency is measured on one query and 100 retrieved documents. Due to the query augmentation, query encoding is more costly and takes about 55ms per query. Even with the augmentation cost, the full dense retrieval total online inference cost is 64ms, only slightly above the BM25 retrieval latency. The ANN retrieval is very efficient, only takes 9ms. In addition, the complexity of ANN retrieval is sublinear to the corpus size, in most ANN framework such as FAISS. Thus the extra round of ANN retrieval operation in MoMA is not the bottleneck even when the size of memory mixture scales up.

## 5.2 Performance with Different Memories

Table 4 evaluates how MoMA behaves under different combinations of external memories. Unsurprisingly, using a single out-of-domain memory for retrieval augmentation does not help, for example, even though MARCO is the source domain corpus, solely grounding on it reduces zero-shot accuracy. MeSH as the sole augmenting corpus also lowers performance, even on some medical retrieval tasks such as BioASQ. Interestingly, when we expand the memory to include MARCO, Wiki, and MeSH, but keep the target corpus excluded (*w/o Target*), MoMA exhibits better accuracy compared to the no-memory version. **Our conclusion is that more memory sources achieves better generalization, especially when no target domain information is available.**

In the *Full* setting, the 3-memory mixture of MARCO, Wiki, and MeSH is jointly learned with final task at training time. At test time, MARCO is swapped out for the target corpus. The *Full* improves zero-shot accuracy over both the *w/o Target* setting (where the target corpus is excluded at test time), and the *w/o Learning* setting (wherein the augmentation component is not learned). As expected, plugging in the target corpus at test time is the most valuable source of generalization power. **It is also the most realistic, as access to the target corpus may only be available at testing time.**

## 5.3 Effect of Memory Mixture Learning

To study the effect of our joint learning mechanism on the memory mixture, we compare it with recent state-of-the-art Attention Distillation (ADist), which is first used in Izacard and Grave (2020a) and recently updated in a parallel work Izacard et al. (2022). It jointly trains the augmentation model using attention scores from the end language model as pseudo-labels. We also enrich ADist with relevance labels from MARCO for more direct supervision (ADist + MSMARCO rel). To exclude the effect of contrastive pretraining, we choose MoMA (T5-ANCE) as our own method for comparison. We also tried using a trained ANCE retriever without further distilling and denote it as w/o Distilling (T5-ANCE). The performances of these joint learning methods are listed in Table 5. The results show that ADist, either standalone or enriched with MARCO labels, does not improve the final accuracy compared to using a supervised dense retriever T5-ANCE. The main difference is

Table 4: NDCG@10 of MoMA under different memory compositions: no memory, single memory, and a mixture of memories. *w/o Learning* uses the end retriever to select augmenting documents without use of an augmentation component. *w/o Target* excludes the target from memory. Best results are in bold.

| | No Memory | Single Memory | | | | Memory Mixture | | |
|---|---|---|---|---|---|---|---|---|
| | | MARCO | Wiki | MeSH | Target | w/o Learning | w/o Target | Full |
| TREC-COVID | 0.653 | 0.576 | 0.592 | 0.669 | 0.731 | 0.759 | 0.664 | **0.761** |
| BioASQ | 0.322 | 0.247 | 0.262 | 0.219 | 0.361 | 0.359 | 0.271 | **0.371** |
| NFCorpus | 0.275 | 0.295 | 0.302 | 0.282 | 0.319 | 0.317 | 0.301 | **0.333** |
| NQ | 0.452 | 0.472 | 0.486 | 0.393 | 0.483 | 0.510 | 0.484 | **0.544** |
| HotpotQA | 0.487 | 0.481 | 0.519 | 0.462 | 0.538 | 0.539 | 0.520 | **0.589** |
| FiQA-2018 | 0.294 | 0.296 | 0.286 | 0.280 | 0.320 | 0.304 | 0.285 | **0.329** |
| Signal-1M | 0.246 | 0.239 | 0.225 | 0.238 | 0.250 | 0.248 | 0.240 | **0.264** |
| TREC-NEWS | 0.379 | 0.381 | 0.391 | 0.372 | 0.416 | 0.410 | 0.398 | **0.453** |
| Robust04 | 0.412 | 0.435 | 0.443 | 0.428 | **0.483** | 0.446 | 0.452 | 0.475 |
| ArguAna | 0.415 | 0.439 | 0.438 | 0.442 | 0.439 | 0.427 | 0.438 | **0.463** |
| Touché-2020 | 0.312 | 0.281 | 0.281 | 0.252 | **0.331** | 0.275 | 0.272 | 0.299 |
| Quora | 0.836 | 0.809 | 0.798 | 0.835 | 0.781 | 0.813 | 0.812 | **0.843** |
| DBPedia-entity | 0.290 | 0.340 | 0.341 | 0.287 | 0.335 | 0.331 | 0.342 | **0.383** |
| SCIDOCS | 0.115 | 0.128 | 0.121 | 0.130 | **0.146** | 0.134 | 0.127 | 0.145 |
| Fever | 0.655 | 0.663 | 0.735 | 0.610 | 0.694 | 0.718 | 0.737 | **0.745** |
| Climate-Fever | 0.194 | 0.231 | 0.238 | 0.231 | 0.228 | 0.222 | **0.240** | 0.233 |
| SciFact | 0.566 | 0.583 | 0.587 | 0.585 | 0.624 | 0.618 | 0.598 | **0.630** |
| CQADupStack | 0.283 | 0.207 | 0.218 | 0.203 | 0.283 | 0.235 | 0.215 | **0.294** |
| Avg | 0.399 | 0.395 | 0.403 | 0.384 | 0.431 | 0.426 | 0.411 | **0.453** |

Table 5: Zero-shot Performances of different distillation methods. We observe consistent trend on all BEIR datasets. We present results on 6 representative datasets from Wikipedia or medical domains.

| Distillation Method | TREC-COVID | BIOASQ | NFCorpus | NQ | HotpotQA | FEVER | **Avg** |
|---|---|---|---|---|---|---|---|
| **Soft Attention Distill** | | | | | | | |
| ADist (Izacard et al., 2022) | 0.609 | 0.185 | 0.227 | 0.351 | 0.387 | 0.615 | 0.396 |
| ADist + MSMARCO rel | 0.664 | 0.220 | 0.255 | 0.397 | 0.394 | 0.624 | 0.426 |
| **w/o Distilling (T5-ANCE)** | 0.741 | 0.361 | 0.301 | 0.472 | 0.513 | 0.684 | 0.512 |
| **MoMA** | **0.762** | **0.372** | **0.307** | **0.490** | **0.539** | 0.723 | **0.532** |

that ADist learns a soft attention score distribution, while the supervised retriever is trained effectively using hard negative sampling (Xiong et al., 2020). Jointly learning using soft labels without hard negatives downgraded the augmentation accuracy. Hence, MoMA is a simple technique to learn the end task signals via the attention scores together with hard negatives, which improves quality over a supervised retriever alone.

To further illustrate the joint training process, we track the attention scores of documents from different memory sources as well as their ratio in the augmentation set in Figure 2. We also split MARCO documents by whether they are labeled as **Relevant (Rel)** for the corresponding query.

Firstly, MoMA learns to increasingly attend to, and retrieve, relevant documents from the memory mixture throughout training. In Figure 2a, more attention is paid to MARCO Relevant documents than to any other type in the memory. Although the number of MARCO Relevant documents is not significant as a percentage of the augmenting set in Figure 2c, a query level analysis confirms that

percentage of queries having at least one relevant document in the augmenting set increases from 46% in Epi-0 to 62% in Epi-2.

This apparent discrepancy can be explained by the fact that MARCO has only one relevant label per query on average, leaving plenty of room for other types of documents to be included in the augmenting set.

Secondly, the amount of attention paid to certain types of documents by MoMA is positively correlated with their representation in the augmenting set. This confirms that the joint learning effectively conveys the feedback signals from the end model to the augmentation component. For instance, in Figure 2a, MoMA pays a high level of attention to MARCO Other documents, a signal reflected in the composition of its augmentation set in Figure 2c. Even though MARCO Other documents were not labeled relevant for the query, they can still prove to be valuable as an augmenting document because they may contain partial information that helps query understanding (Lavrenko and Croft, 2017) or it was simply not annotated

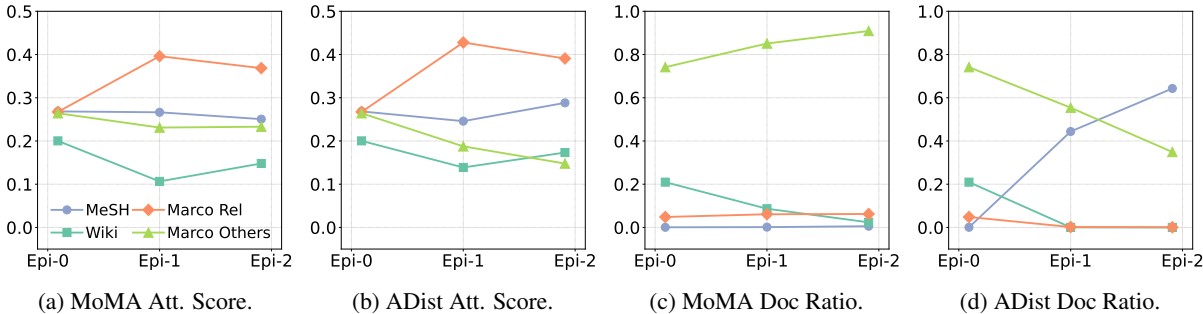

Figure 2: Grounding component breakdown for different distillation methods in each learning iteration. We display the regularized doc and att. score ratio of documents from different augmentation sources.

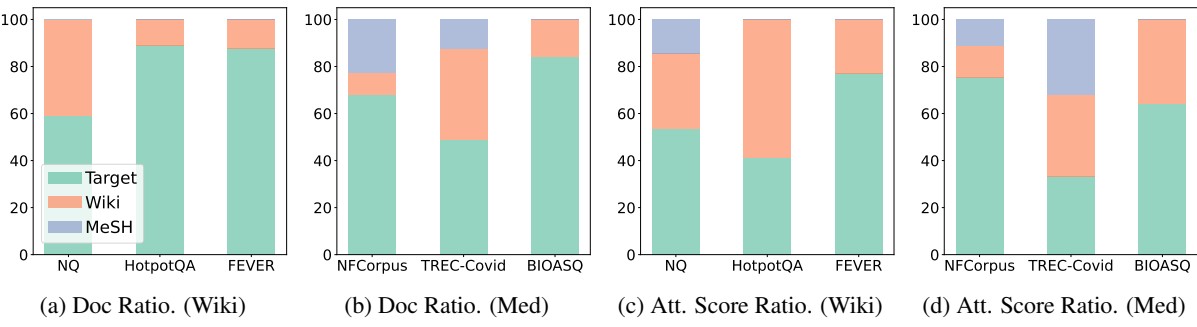

Figure 3: The inclusion of Plug-In memory during testing (grouped by the Wiki and Medical domains).

in MARCO's sparse labels (Bajaj et al., 2016). In comparison, the correlation of the two in ADist is weak as the model seems to include 60% augmenting documents from MeSH, far greater than the fraction of medical queries in MARCO.

## 5.4 Generalization of Plug-In Memory

In the previous section, we observed how MoMA learns to attend to, and retrieve, informative documents from memories on which it was trained. In this section, we examine the zero-shot behavior of MoMA (T5-ANCE) on new corpora plugged-in at test time (keeping Wiki and MeSH as before).

Figure 3 compares documents from the plugged-in target versus the remaining memory mixture in terms of membership in the augmenting set (Doc Ratio) and attention. Again, on all tasks, MoMA (T5-ANCE) heavily attends to – and successfully retrieves – in-domain documents, even if those in-domain documents were only just plugged in. This confirms that the augmentation model achieves the zero-shot ability to capture relevant information from unseen corpora.

In the medical domain, the model pays more attention to MeSH documents, especially on TREC-Covid task since MeSH includes high quality updated information related to COVID-19. Wikipedia documents received more attention on the Wiki-

centric tasks like FEVER, as expected. Some tasks may need a small amount of precise information from Wikipedia to answer the detailed question, e.g. in HotpotQA. Similar with the training process, there is a non-trivial correspondence between attention score of a memory and its membership in the augmentation set.

## 6 Conclusion

In this paper we propose a new plug-in mixture-of-memory mechanism for the retrieval augmented language models to improve their zero-shot ability on the dense retrieval task. To learn the memory mixture we develop a new joint learning approach that trains the augmentation component using the positive signals from the end task, the language model's attention scores, and hard negatives retrieved from the mixture of augmentation corpora. This leads to our final model MoMA (T5-ANCE) and MoMA (coCondenser) that achieve strong zero-shot accuracy on 18 retrieval tasks included in BEIR. Our analysis shows the importance of augmenting with diverse memory sources and in-domain information for robust generalization. We hope our findings can inspire more future research in better augmenting language models, to provide other alternatives to achieve generalization ability beyond solely relying on model scale.

## Limitations

Although MoMA (T5-ANCE) and MoMA (coCondenser) achieve strong zero-shot performances, we mainly verify their efficacy from the empirical performances on BEIR tasks, where the target corpora, Wiki and MARCO serve as readily available retrieval sources. In a real-world scenario, the grounding corpora usually need to be customized according to query domains and user needs. Thus, how to choose effective grounding corpora and efficiently evaluate their relative contribution remain an open problem. These analyses will go beyond our empirical settings and reveal a wider application scenario of MoMA.

## Ethics Statement

All data in this study are publicly available and used under ethical considerations. Text and figures in the paper are used for illustration only, they do not represent the ethical attitude of the authors.

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

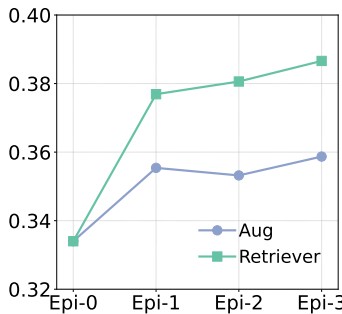

Figure 4: MRR@10 on MSMARCO of the augmentation component and end retriever during MoMA (T5-ANCE) training.

# A  Appendix

## A.1  Performance on Source Domain

Figure 4 demonstrates the MRR@10 on MS-MARCO for the end retriever and augmentation component of MoMA (coCondenser) over different training epochs. We make the following observations. Firstly, the augmentation component improves on the source domain even though it is not directly optimized with relevance labels. Since helpful augmentation documents are usually strongly related to the query, the augmentation component benefits from such indirect relevance signals. Secondly, the end retriever monotonically benefits from information collected by the augmenting component, indicating that the two components mutually enhance each other in the joint learning process. We further compare MoMA with relevant baselines on MSMARCO in Table 6. The comparison verifies that MoMA also achieves better performance on the source domain retrieval tasks.

## A.2  Case Studies

Table 7 shows examples of how augmenting documents chosen by MoMA can provide valuable contextual information for the query. The first example is a training query from MARCO, where the augmenting documents help disambiguate the query word "rating". In the second one, documents from the official Wiki and HotpotQA's Wiki corpus are descriptions of the two entities in HotpotQA's comparison question. It illustrates how MoMA provides more comprehensive augmentation by incorporating information from different sources. The last query shows the benefit of the in-domain plug-in corpus as it brings in very specific information about the query (AND-1/Ctf4) that is hard to find elsewhere.

## A.3  Datasets Details

**Evaluation Datasets**   Target domain datasets used in our experiments are collected in the BEIR benchmark (Thakur et al., 2021b)[4] and include the following domains:

- Open-domain Question Answering (QA): HotpotQA (Yang et al., 2018), NQ (Kwiatkowski et al., 2019), and FiQA (Maia et al., 2018).

- Bio-Medical Information Retrieval: TREC-COVID (Voorhees et al., 2021), NFCorpus (Boteva et al., 2016), and BioASQ (Tsatsaronis et al., 2015).

- Argument Retrieval: Webis-Touché2020 (Bondarenko et al., 2020) and ArguAna (Wachsmuth et al., 2018).

- News Retrieval: TREC-NEWS (Soboroff et al., 2018) and Robust04 (Voorhees et al., 2004).

- Tweet Retrieval: Signal-1m (Suarez et al., 2018).

- Duplicate Question Retrieval: Quora (Thakur et al., 2021b) and CQADupStack (Hoogeveen et al., 2015).

- Entity Retrieval: DBPedia (Hasibi et al., 2017)

- Citation Prediction: SCIDOCS (Cohan et al., 2020)

- Fact Checking: SciFact (Wadden et al., 2020), FEVER (Thorne et al., 2018), and Climate-FEVER (Diggelmann et al., 2020)

We list the statistics of the BEIR benchmark in Table 8.

**Augmenting Corpora**   Corpus size We first introduce more details on how we preprocessed the Medical Subject Headings (MeSH) Database. We select text information from the Qualifier Record Set and Descriptor Record Set. Each set contains multiple <Concept> elements, which is composed of three sub-elecments, i.e., <ConceptName>, <ScopeNote> and <TermList>. Among the sub-elecments, <ScopeNote> is the major textual information source, which is usually a short description to a medical term or phenomenon. We directly consider each <ScopeNote> as a document entry and concatenate it with corresponding <ConceptName>.

We list the statistics of the augmenting corpora in Table 9.

---

[4] https://github.com/beir-cellar/beir

Table 6: Performance comparisons of different methods on MSMARCO.

|  | DPR | T5-ANCE | coCondenser | MoMA (T5-ANCE) | MoMA (coCondenser) |
|---|---|---|---|---|---|
| MRR@10 | 0.3340 | 0.3678 | 0.3820 | 0.3866 | 0.4056 |

Table 7: MoMA retrieves augmenting documents during training (Marco) and testing (BEIR).

| Queries | Augmentation Docs |
|---|---|
| **Training** | |
| **[Marco]** What is hotel transylvania rated | **[Marco]** Why is Hotel Transylvania 2 rated PG? It is rated PG for some scary images, action and rude humor. **[Wiki]** Another review aggregate calculated an average score of 47 out of 100, indicating "mixed or average reviews". |
| **Zero-Shot Testing** | |
| **[HotpotQA]** Were Scott Derrickson and Ed Wood of the same nationality? | **[Wiki]** Scott Derrickson (born July 16, 1966) is an American director, screenwriter and producer. **[HotpotQA]** Edward Davis Wood Jr. (October 10, December 10, 1978) was an American filmmaker, actor, writer, producer, and director. |
| **[BIOASQ]** Is AND-1/Ctf4 essential for proliferation? | **[BIOASQ]** AND-1/Ctf4 bridges the CMG helicase and DNA polymerase alpha, facilitating replication. **[Wiki]** FADD has no effect on the proliferation of B cells induced by stimulation of the B cell receptor. |

## A.4 Baselines

We use the baselines from the current BEIR leaderboard (Thakur et al., 2021b) and recent papers. These baselines can be divided into four groups: dense retrieval, dense retrieval with generated queries[5], lexical retrieval and late interaction.

**Dense Retrieval** For dense retrieval, the baselines are the same dual-tower model as ours. We consider **DPR** (Karpukhin et al., 2020), **ANCE** (Xiong et al., 2020), **T5-ANCE**, **coCondenser** (Gao and Callan, 2022) and one recently-proposed model **GTR** (Ni et al., 2021) with different size configuration in this paper.

- **DPR** uses a single BM25 retrieval example and in-batch examples as hard negative examples to train the model. Different from the original paper (Thakur et al., 2021b) that train the DPR on QA datasets, we train DPR on MS MARCO (Bajaj et al., 2016) Dataset for *fair comparison*. Notice that this also lead to better results according to Xin et al. (2022).

- **ANCE** constructs hard negative examples from an ANN index of the corpus. The hard negative training instances are updated in parallel during fine-tuning of the model. The model is a RoBERTa (Liu et al., 2019) model trained on MS MARCO for 600k steps.

- **T5-ANCE** Different with default ANCE setting, we replace the backbone language model RoBERTa with T5-base. All the other model settings are the same with the original ANCE. We include this baseline because as a subroutine for MoMA, it could be viewed as an ablation without memory augmentation. We can directly observe the impact of plug-in mixture of memory by comparing T5-ANCE with MoMA.

- **coCondenser** is a continuous pre-trained model based on BERT, with the equivalent amount of parameters to BERT-base. It enhances the representation ability of [CLS] token by changing the connections between different layers of Transformer blocks. Fine-tuning of coCondenser uses BM25 and self-mined negatives.

- **Contriever** conducts unsupervised contrastive pretraining with data augmentations and momentum queues on Wikipedia and the larger CC-Net (Wenzek et al., 2020) corpora for 500k steps.

- **GTR** initializes the dual encoders from the T5 models (Raffel et al., 2019). It is first pre-trained on Community QA[6] with 2 billion question-answer pairs then fine-tuned on NQ and MS Marco dataset. In addition, they use the hard negatives released by RocketQA (Qu et al., 2021) when finetuning with MS Marco data and the hard negatives release by (Lu et al., 2021) for Natural Questions. **GTR_base** leverages the same T5-base model as MoMA, while **GTR_large** is based on T5-large, which is not directly comparable to our method as it triples the parameters.

**Dense Retrieval with Generated Queries** **GenQ** first fine-tunes a T5-base (Raffel et al., 2019)

---

[5]We separate them from dense retrieval since they usually rely on Seq2seq models to generate pseudo query-document pairs, and they train a model for each dataset *independently* instead of using a single model for all datasets.

[6]Unfortunately, this corpus has not been released by the authors.

Table 8: Statistics of datasets in the BEIR benchmark. The table is taken from the original BEIR benchmark paper (Thakur et al., 2021b).

| Split (→) Task (↓) | Domain (↓) | Dataset (↓) | Title | Relevancy | Train #Pairs | Dev #Query | Test #Query | Test #Corpus | Test Avg. D / Q | Avg. Word Lengths Query | Avg. Word Lengths Document |
|---|---|---|---|---|---|---|---|---|---|---|---|
| Passage-Retrieval | Misc. | MS MARCO | ✗ | Binary | 532,761 | —— | 6,980 | 8,841,823 | 1.1 | 5.96 | 55.98 |
| Bio-Medical | Bio-Medical | TREC-COVID | ✓ | 3-level | —— | —— | 50 | 171,332 | 493.5 | 10.60 | 160.77 |
| Information | Bio-Medical | NFCorpus | ✓ | 3-level | 110,575 | 324 | 323 | 3,633 | 38.2 | 3.30 | 232.26 |
| Retrieval (IR) | Bio-Medical | BioASQ | ✓ | Binary | 32,916 | —— | 500 | 14,914,602 | 4.7 | 8.05 | 202.61 |
| Question | Wikipedia | NQ | ✓ | Binary | 132,803 | —— | 3,452 | 2,681,468 | 1.2 | 9.16 | 78.88 |
| Answering | Wikipedia | HotpotQA | ✓ | Binary | 170,000 | 5,447 | 7,405 | 5,233,329 | 2.0 | 17.61 | 46.30 |
| (QA) | Finance | FiQA-2018 | ✗ | Binary | 14,166 | 500 | 648 | 57,638 | 2.6 | 10.77 | 132.32 |
| Tweet-Retrieval | Twitter | Signal-1M (RT) | ✗ | 3-level | —— | —— | 97 | 2,866,316 | 19.6 | 9.30 | 13.93 |
| News | News | TREC-NEWS | ✓ | 5-level | —— | —— | 57 | 594,977 | 19.6 | 11.14 | 634.79 |
| Retrieval | News | Robust04 | ✗ | 3-level | —— | —— | 249 | 528,155 | 69.9 | 15.27 | 466.40 |
| Argument | Misc. | ArguAna | ✓ | Binary | —— | —— | 1,406 | 8,674 | 1.0 | 192.98 | 166.80 |
| Retrieval | Misc. | Touché-2020 | ✓ | 3-level | —— | —— | 49 | 382,545 | 19.0 | 6.55 | 292.37 |
| Duplicate-Question | StackEx. | CQADupStack | ✓ | Binary | —— | —— | 13,145 | 457,199 | 1.4 | 8.59 | 129.09 |
| Retrieval | Quora | Quora | ✗ | Binary | —— | 5,000 | 10,000 | 522,931 | 1.6 | 9.53 | 11.44 |
| Entity-Retrieval | Wikipedia | DBPedia | ✓ | 3-level | —— | 67 | 400 | 4,635,922 | 38.2 | 5.39 | 49.68 |
| Citation-Prediction | Scientific | SCIDOCS | ✓ | Binary | —— | —— | 1,000 | 25,657 | 4.9 | 9.38 | 176.19 |
| Fact Checking | Wikipedia | FEVER | ✓ | Binary | 140,085 | 6,666 | 6,666 | 5,416,568 | 1.2 | 8.13 | 84.76 |
| | Wikipedia | Climate-FEVER | ✓ | Binary | —— | —— | 1,535 | 5,416,593 | 3.0 | 20.13 | 84.76 |
| | Scientific | SciFact | ✓ | Binary | 920 | —— | 300 | 5,183 | 1.1 | 12.37 | 213.63 |

Table 9: Statistics of the augmenting corpora.

| Datasets | Corpus Size | Avg. Doc Length |
|---|---|---|
| MS MARCO | 502,939 | 56.0 |
| MeSH | 32,326 | 16.8 |
| Wiki | 21,015,324 | 100.0 |

model on MS MARCO for 2 epochs and then generate 5 queries for each passage as additional training data for the target domain to continue to fine-tune the TAS-B (Hofstätter et al., 2021) model.

**Lexical Retrieval** Lexical retrieval is a score function for token matching calculated between two high-dimensional sparse vectors with token weights. **BM25** (Robertson et al., 2009) is the most commonly used lexical retrieval function. We use the BM25 results reported in Thakur et al. (2021b) for comparison.

**Late Interaction** We also consider a late interaction baseline, namely **ColBERT** (Khattab and Zaharia, 2020b). The model computes multiple contextualized embeddings for each token of queries and documents, and then uses a maximum similarity function to retrieve relevant documents. This type of matching requires significantly more disk space for indexes and has a higher latency.

## A.5 Detailed Experimental Settings and hyperparameters

Our implementation uses PyTorch (Paszke et al., 2019) with Hugging Face Transformers (Wolf et al., 2020). We optimize the model using

AdamW (Loshchilov and Hutter, 2019) with a peak learning rate at 5e-6, weight decay of 0.01, and linear learning rate decay. The global batch size is set to 256. The maximum length of query and passage are set to 32 and 128 respectively. We summarize all hyperparameter settings in Table 10. The model is trained with 8 Nvidia A100 80GB GPUs and FP16 mixed-precision training. The total running time is 6.6 hrs for three episodes of augmentation component training and 6.3 hrs for end retriever training. We detail the training time of each episode in Table 11.

When evaluating on the BEIR benchmark, we follow the setting in GTR (Ni et al., 2021), which use sequences of 64 tokens for the questions and 512 for the documents in all datasets except Trec-News, Robust-04 and ArguAna. In particular, we set the document length to 768 for Trec-News and Robust-04. For ArguAna, we set both question and document length to 128. The above length setting is in accordance to the average query and document lengths in these datasets.

Table 10: The hyperparameters of MoMA.

| Hyperparameters | Settings |
|---|---|
| Grounding document number | 10 |
| Attention threshold number | 5 |
| Negative mining depth | 200 |
| Global batch size (query size per batch) | 256 |
| Positive number per query | 1 |
| Negative number per query | 7 |
| Peak learnig rate | 5e-6 |
| Learnig rate decay | 0.01 |
| Optimizer | AdamW |
| Scheduler | Linear |
| MARCO Maximum query length | 32 |
| MARCO Maximum document length | 128 |

Table 11: Training time for MoMA with three training episodes. We use 8 Nvidia A100 80GB GPUs with FP16 mixed-precision training.

| Stage | Augmentation Component | End Retriever |
|---|---|---|
| Epi-1 | 0.8h | 1.5h |
| Epi-2 | 0.8h | 1.5h |
| Epi-3 | 0.8h | 1.5h |
| Index refresh | 1.4h | 0.6h |
| Refresh number | 3 | 3 |
| Overall | 6.6h | 6.3h |