# OpenReview forum: "Augmenting Zero-Shot Dense Retrievers with Plug-in Mixture-of-Memories"
_EMNLP/2023/Conference — EMNLP 2023 Main_

### Official Review · Reviewer_i6oZ · 2023-08-05

**Soundness:** 4

**Ethical Concerns:**

Yes

**Excitement:**

4: Strong: This paper deepens the understanding of some phenomenon or lowers the barriers to an existing research direction.

**Missing References:**

Hierarchical Memory Networks: https://arxiv.org/pdf/1605.07427.pdf  -- this work might be worth mentioning as this is one of the earlier works that attempts to retrieve K memory specially the clustering variants proposed in this paper.

**Paper Topic And Main Contributions:**

This paper presents Mixture-Of-Memory Augmentation (MoMA), a method to improve the zero-shot generalization ability of large language models (LLMs). MoMA retrieves augmentation documents from multiple information corpora, which we refer to as "external memories." This is inspired by query expansion work in the information retrieval (IR) literature. The method allows us to update the corpora, which in turn allows us to update the memory of LLMs without parameter tuning.
This work presents learning mechanism that trains the augmentation component. Authors evaluate the results on several tasks from the BEIR benchmark. Experiments and results suggest that the proposed MoMA approach is superior to other dense retrieval approaches

**Questions For The Authors:**

- Is it possible to setup a dense retrieval baseline that uses ST5-EncDec variant of Sentence-T5 and retried variants are restricted to come from different sources? It would require encoder-decoder architecture to consumer information from all the sources though.
- Are the results in Table 1 averaged over different trails assuming that they might be vary on different trials?


**Reasons To Accept:**

- This work presents a clear and well-motivated technique to address the problem zero-shot document augmentation retrievers.
- This method shows improved results consistently across several benchmarks. Additionally, effect of memory mixture is presented empirically which is both effective and show strong generalization performance new domains.
- The very 'plug-n-play' nature of the proposed approach makes it widely applicable therefore is of interest for the NLP research community.

**Reasons To Reject:**

Please see questions and respond to them.

**Reproducibility:**

4: Could mostly reproduce the results, but there may be some variation because of sample variance or minor variations in their interpretation of the protocol or method.

**Reviewer Confidence:**

3: Pretty sure, but there's a chance I missed something. Although I have a good feel for this area in general, I did not carefully check the paper's details, e.g., the math, experimental design, or novelty.

---

> ### Author Rebuttal · Authors · 2023-08-28
>
> Thank you for the useful questions and suggestions!
>
> Q1: Is it possible to set up a dense retrieval baseline that uses ST5-EncDec variant of Sentence-T5 and retried variants are restricted to come from different sources? It would require encoder-decoder architecture to consume information from all the sources though.
>
> A1: We agreed that such a baseline would be helpful. In the paper, we combine corpus from different sources into the same ANN index, since manually controlling the ratio of different sources may impose some inductive bias. In the next version, we will add a baseline that manually restricts the data source. We also want to mention that our T5-ANCE used the architecture of the encoder-decoder variant of Sentence-T5, as we take the decoder output as query representation. We will make this description in Section 3.1 more explicit. Overall, T5-ANCE is a stronger (improved) ST5-EncDec baseline since it is trained with negative labels on MSMARCO.
>
> Q2: Are the results in Table 1 averaged over different trails assuming that they might vary on different trials?
>
> A2: Results may vary across different runs, but the improvement of MoMA over Contriever and $GTR_{large} is statistically significant with p-value < 0.05. The improvement of MoMA over other baselines is significant with p-value < 0.01. We will add the significance test in the next version. Also, we ensure that all our runs are random and we avoid cherry-picking. From our experience, MoMA and T5-ANCE, etc. are stable as long as the hyperparameters are set in a reasonable range.
>
> Missing Reference:
>
> Thanks for the suggestion. It is an insightful observation of the connection between memory networks and memory augmentation in MoMA. We will add it to the reference and discuss it in the related work as well.

---

### Official Review · Reviewer_3HsD · 2023-08-12

**Soundness:** 3

**Excitement:**

3: Ambivalent: It has merits (e.g., it reports state-of-the-art results, the idea is nice), but there are key weaknesses (e.g., it describes incremental work), and it can significantly benefit from another round of revision. However, I won't object to accepting it if my co-reviewers champion it.

**Paper Topic And Main Contributions:**

This paper introduces a new method for augmenting a dense retriever. The proposed method first retrives relevant documents from an external collection and then produces the query's dense representation conditioned on the original query and the retrieved documents. The authors tested the method on BEIR. The method performs better than some other dense retrival methods and generalizes better out of distribution.

**Questions For The Authors:**

1. How is the in-domain result on MSMARCO?
2. How is the trained augmentation component $f^a()$ performs alone on retrieval?

**Reasons To Accept:**

The purpose method seems novel and the results are positive. The method generalizes well to out-of-domain data, which is a common problem for other dense retrieval methods. The paper is well-written with detailed ablation study and discussion.

**Reasons To Reject:**

The ablation study shows that most performance gain comes from the inclusion of the target corpus. When the target corpus is excluded, the performance improvement is minimal or even reversed. This result brings several concerns:
1. Since the target corpus is retrieved once and is fused later, this framework resembles a retrieve and re-rank schema. How does this method compare to a retrieve and re-rank method?
2. Since the inclusion of the target corpus is crucial to the performance, the purpose method could be inefficient when the target corpus is very large.

**Reproducibility:**

4: Could mostly reproduce the results, but there may be some variation because of sample variance or minor variations in their interpretation of the protocol or method.

**Reviewer Confidence:**

3: Pretty sure, but there's a chance I missed something. Although I have a good feel for this area in general, I did not carefully check the paper's details, e.g., the math, experimental design, or novelty.

---

> ### Author Rebuttal · Authors · 2023-08-28
>
> Thank you for the useful questions and suggestions!
>
> W1: When the target corpus is excluded, the performance improvement is minimal or even reversed.
>
> A1: We will first explain the reason for the minimal performance improvement and then suggest ways to improve.
> The marginal improvement on a few datasets could be traced to the semantic disparities between augmentation corpus and target queries. Since we use a fixed set of memory for all target tasks, it may not be as effective when the task differs significantly from the augmentation corpus (MSMARCO and Wikipedia) in terms of topics and styles. For example in Table 1, MoMA performs well on queries asking about general world knowledge (NQ, DBPedia, and Robust04), which are related to or can be answered by MSMARCO or Wikipedia in our memory mixture. However, when the queries differ a lot in topic, such as tweet retrieval for a corpus (Signal-1M), the improvement from MoMA is marginal. The observation coincides with the results in Table 4, which we summarize here as:
>
> |Task|No Memory|+MARCO|+Wiki|+Target|Mixture w/o Target|Mixture w Target|
> |:---:|:---:|:---:|:---:|:---:|:---:|:---:|
> |NQ|0.452|0.472|0.486|0.483|0.484|0.544|
> |DBPedia|0.290|0.340|0.341|0.335|0.342|0.383|
> |Robust04|0.412|0.435|0.443|0.483|0.452|0.475|
> |Signal-1M|0.246|0.239|0.225|0.250|0.240|0.264|
>
> On the first three tasks, using each memory alone already brings a substantial improvement, and together they lead to a slightly better performance. However, on Signal-1M, MoMA is not as effective when each individual memory is not helpful enough. An intuitive explanation could be if the external corpus is very different from the usage scenario, then there is not much beneficial information from there. To conclude, finding an external corpus that includes useful information for the task is important for MoMA, and perhaps for most retrieval-augmented systems. To ensure a stable performance improvement, we suggest 1) using a relevant corpus that matches with the query forms and topics, e.g., the target corpus. In fact, the target corpus is available for the plug-in in many applications, making MoMA feasible for real-world scenarios, or 2) using an up-to-date and comprehensive corpus e.g., the real-world web search scenario. We will add the analysis and discussion in our next version.
>
> W2: Since the target corpus is retrieved once and is fused later, how does this method compare to a retrieve and re-rank method?
>
> A2: 1) The major reason of MoMA being a retrieval model not re-ranking is its design and application. Retrieval is to find top-K from the entire corpus where more efficiency restrictions are needed, e.g., dual-encoder where documents are encoded independently with the query. Reranking is to further rankthe top retrieved results. Thus more expensive models can be used, such as cross-encoders. However, reranking has to be done after an initial retrieval. MoMA belongs to the first stage retrieval as it is to find top-K documents from the entire corpus and it is a dual-encoder model. Any reranking model can be applied after MoMA.
> 2) Another minor difference is how they use the augmentation documents. In MoMA, the top retrieved documents may not overlap with augmentation documents. Augmentation documents are used as input together with the original query to learn a better dense vector representation. We then use the dense vector to retrieve from the entire document set. While in re-ranking, the final results are a shuffled list of first-round retrieved documents.
>
> W3: Since the inclusion of the target corpus is crucial to the performance, the purpose method could be inefficient when the target corpus is very large.
>
> A3: From Table 3, we can decompose the computation cost of MoMA into three parts:
> 1) Encoding of target corpus: Since dense retrieval models need to retrieve from the target, it is an unavoidable computation cost for any model. Moreover, it can be computed offline.
> 2) Query Encoding: 55ms, which is the major online computation cost of MoMA. However, it is a cost irrelevant to the target corpus size, as we keep a fixed number of K (K=10) of augmentation documents. During query encoding, we only encode the target query along with K augmented documents, so the computation is stable for different corpus sizes.
> 3) ANN Retrieval: 9ms as the sum of two retrieval rounds, which is an additional but moderate cost of MoMA. Compared with other dense retrieval models, MoMA needs to retrieve twice, bringing an overhead of ~4.5ms. This accounts for at most 4.5/(9+55)=7.03% time cost. The cost may slightly grow when the target corpus is large, but LSH-based ANN index systems in general can achieve sub-linear complexity regarding the corpus size when needed [1].
>
> Q1&2: How is the in-domain result on MSMARCO? How does the trained augmentation component fa() perform alone on retrieval?
>
> A1&2:
> ||Epi-0|Epi-1|Epi-2|Epi-3|
> |:---:|:---:|:---:|:---:|:---:|
> |Retriever|0.3520|0.3818|0.3929|0.4056|
> |Augmentation Component|0.3520|0.3598|0.3554|0.3643|
>
> The table demonstrates the MRR@10 on MSMARCO for the end retriever and augmentation component of MoMA (coCondenser) over different training epochs.We make the following observations. Firstly, the augmentation component improves on the source domain even though it is not directly optimized with relevance labels. Since helpful augmentation documents are usually strongly related to the query, the augmentation component benefits from such indirect relevance signals. Secondly, the end retriever monotonically benefits from information collected by the augmenting component, indicating that the two components mutually enhance each other in the joint learning process. We further compare MoMA with relevant baselines on MSMARCO:
>
> ||DPR|T5-ANCE|coCondenser|MoMA (T5-ANCE)|MoMA (coCondenser)|
> |:---:|:---:|:---:|:---:|:---:|:---:|
> |MRR@10|0.3340|0.3678|0.3820|0.3866|0.4056|
>
> The comparison verifies that MoMA also achieves better performance on the source domain retrieval tasks.
>
> [1] Aumüller et al. ANN-Benchmarks: A Benchmarking Tool for Approximate Nearest Neighbor Algorithms.

---

### Official Review · Reviewer_uvZi · 2023-08-18

**Typos Grammar Style And Presentation Improvements:** NA
**Soundness:** 4

**Excitement:**

4: Strong: This paper deepens the understanding of some phenomenon or lowers the barriers to an existing research direction.

**Missing References:**

NA

**Paper Topic And Main Contributions:**

In this paper, the authors propose a new retrieval augmentation mechanism for dense retriever. The proposed method is simple and effective, built upon knowledge from multiple external corpora. From what I understand, two ingredients that make this model work better than the previous models: (1) using MoMA to provide richer and more diverse external resources for augmentation and (2) jointly training the augmentation component and dense retriever using supervised relevance signals and self-mined hard negatives.

**Questions For The Authors:**

1. The author needs to clarify whether the information retrieval task in this paper is supervised or unsupervised make me confused, e.g. the model training section mentions supervised labels.
2. The author needs to clarify the reasons behind the poor performance of the multiple knowledge enriched model on these datasets. The MoMA (coCondenser) performs worse than BM25 on certain datasets, such as Signal-1M and SciFact. However, the paper does not provide an analysis for this phenomenon.


**Reasons To Accept:**

1. Simple, well-motivated method that works well, and the code will be publicly available.
2. The model is evaluated extensively with good empirical results on IR benchmark (BEIR).
3. The authors conduct sufficient ablation study and verify the effectiveness of the two main ideas in this paper.
4. The proposed method is more efficient than the previous retrieval pretrained models.


**Reasons To Reject:**

1. In fact, whether the information retrieval task in this paper is supervised or unsupervised make me confused, e.g. the model training section mentions supervised labels. It would be better to clarify in the paper.
2. MoMA (coCondenser) performs worse than BM25 on certain datasets, such as Signal-1M and SciFact. However, the paper does not provide an analysis for this phenomenon. If possible, I would like to know the reasons behind the poor performance of the multiple knowledge enriched model on these datasets.


**Reproducibility:**

4: Could mostly reproduce the results, but there may be some variation because of sample variance or minor variations in their interpretation of the protocol or method.

**Reviewer Confidence:**

3: Pretty sure, but there's a chance I missed something. Although I have a good feel for this area in general, I did not carefully check the paper's details, e.g., the math, experimental design, or novelty.

---

> ### Author Rebuttal · Authors · 2023-08-28
>
> Thank you for the useful questions and suggestions!
>
> W1: In fact, whether the information retrieval task in this paper is supervised or unsupervised make me confused, e.g. the model training section mentions supervised labels.
>
> A1: The retrieval task is a zero-shot setting where labels are from the source task, and the model is evaluated on target tasks where no labels are available. In our setting, the model is supervised trained on MAMARCO, and zero-shot evaluated on the BEIR benchmark. It is a standard setup in zero-shot retrieval and we follow other baselines [0,1,2] to do so.
>
> W2: MoMA (coCondenser) performs worse than BM25 on certain datasets, such as Signal-1M and SciFact. However, the paper does not provide an analysis for this phenomenon.
>
> A2: The performance downgrade can be attributed to the language style (format and length) disparities between the training (MSMARCO) and evaluation (Signal-1M and SciFact) sets. In fact, results from Table 1 suggest that almost all dense retrieval models without training on external corpus experience the downgrade. For example, ANCE, coCondenser, and GTR. The training corpus for all the models is MSMARCO, a general domain QA corpus originated from the Bing search engine. In contrast, Signal-1M centered on tweet retrieval for news articles, and SciFact is for veracity of scientific claims. Both datasets are significantly different from MSMARCO in terms of length and style. In this case, a traditional lexical match such as BM25 is more robust in this scenario.
>
> [0] Yu et al. Coco-dr: Combating distribution shifts in zero-shot dense retrieval with contrastive and distributionally robust learning. \
> [1] Wang et al. GPL: Generative pseudo labeling for unsupervised domain adaptation of dense retrieval. \
> [2] Ni el al. Large Dual Encoders Are Generalizable Retrievers.

---

### Meta-Review · Area_Chair_oKtg · 2023-09-27

**Recommendation:** 4

**Metareview:**

This paper proposes a new retrieval augmentation mechanism to improve the zero-shot generalization ability of language models. Most of reviewers have given positive scores, indicating that they appreciate the authors' work. Although there are specific concerns raised by each reviewer that need to be addressed for further improvement, the authors have adequately addressed the concerns raised by the reviewers, demonstrating their capability to further improve the paper. The authors should carefully consider and incorporate these suggestions to further enhance the quality of the work.

---

### Decision · Program_Chairs · 2023-10-07

**Decision:**

Accept-Main

**Comment:**

This paper proposes a new retrieval augmentation mechanism to improve the zero-shot generalization ability of language models. Most of reviewers have given positive scores, indicating that they appreciate the authors' work. Although there are specific concerns raised by each reviewer that need to be addressed for further improvement, the authors have adequately addressed the concerns raised by the reviewers, demonstrating their capability to further improve the paper. The authors should carefully consider and incorporate these suggestions to further enhance the quality of the work.